# BUILT-IN SAFEGUARDS FOR AUTO-REGRESSIVE PROTEIN FOUNDATION MODELS THROUGH SELF-PLAY

**Xiaoyi Fu**
Hong Kong University of Science and Technology
Clear Water Bay, Hong Kong SAR, China
`xfu854@connect.ust.hk`

**Yang Xu**
University of Washington
Seattle, WA 98195, United States
`yxu59@uw.edu`

**King L. Chow**
Hong Kong University of Science and Technology
Clear Water Bay, Hong Kong SAR, China
`bokchow@ust.hk`

**Yuan Yao**
Hong Kong University of Science and Technology
Clear Water Bay, Hong Kong SAR, China
`yuany@ust.hk`

## ABSTRACT

The biorisk of dual use is rising for protein generative models as such tools proliferate. Recent work in the machine learning community has introduced frameworks for systematically red-teaming protein foundation models to uncover these risks. However, existing frameworks focus primarily on diffusion models, leaving a gap for autoregressive models that generate protein sequences one amino acid at a time, conditioned on a partial sequence or structure. To address this, we extend the current framework to autoregressive protein generative models and propose a built-in defensive strategy based on model Self-Play. Empirical results on the SafeProtein benchmark show that a GRPO-based method significantly outperforms a standard supervised fine-tuning baseline and exhibits a scaling law for model Self-play. We further examine the impact on generation quality through extensive experiments on two exemplar enzymes.

## 1 INTRODUCTION

Exponential advances in DNA synthesis, spanning cost, quality, and throughput, have greatly simplified the encoding of novel proteins into synthetic genes, while simultaneously exposing underappreciated dual-use biorisks (Baker & Church, 2024). DNA sequence screening prior to synthesis remains a central biosecurity safeguard, currently performed voluntarily by members of the International Gene Synthesis Consortium (IGSC) (Hoffmann et al., 2023). However, dual-use concerns are to be mitigated: first, screening is not mandated and therefore lacks universal coverage (Baker & Church, 2024); second, preventive approaches such as built-in safeguards have been shown to reduce escape frequencies when deployed in conjunction with post-screening strategies (Hoffmann et al., 2023).

### 1.1 PROTEIN FOUNDATION MODELS

The convergence of modern generative modeling paradigms has significantly expanded the frontier of enzyme design, including diffusion models (Ahern et al., 2025), large protein language models (Hayes et al., 2025), and flow-matching architectures (Geffner et al., 2025). Protein language models fine-tuned for functional objectives have emerged as particularly flexible platforms. In parallel, new steering methodologies have matured, ranging from derivative-free guidance (Li et al., 2024) and reinforcement learning approaches (Stocco et al., 2024) to experiment-informed feedback loops (Yang et al., 2025), the latter exemplifying what has recently been termed *experiment-guided steering*.[1]

---

[1] `https://www.nabla.bio/news/test-time-scaling-de-novo-antibodies`

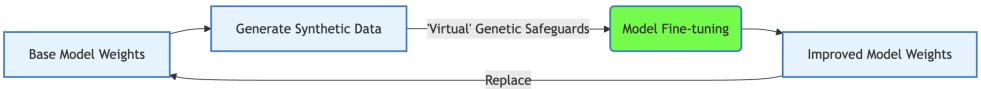

Figure 1: Overview of the proposed Built-in Safeguard through **Self-play**.

## 1.2 BUILT-IN GENETIC SAFEGUARDS

Genetic safeguards are organism-level systems engineered to restrict the environmental or genomic conditions under which synthetic organisms can survive. As detailed in (Hoffmann et al., 2023), enabling techniques span four main categories: gene knockouts that generate auxotrophies, de-repression of suicide genes, control of essential gene function (Cai et al., 2015), and orthogonal central dogma architectures (Agmon et al., 2017). In practice, multilayered safeguard systems (Gallagher et al., 2015) can further strengthen biocontainment and support the safe deployment of engineered circuits in clinical and industrial settings.

Analogous to these 'physical' safeguards, 'virtual' built-in safeguards for generative protein AI systems are emerging (Wang et al., 2025a). Proposed methods include watermarking (Zhang et al., 2024), model safety alignment (Wang et al., 2025a), unlearning techniques (Bourtoule et al., 2021), and anti-jailbreak defenses (Zhang et al., 2025). However, these approaches remain early in development and are not yet widely adopted or fully operational.

**Contributions** This work presents a novel framework of Built-in Safeguards for auto-regressive protein foundation models, and our contributions are threefold:

- We extend the SafeProtein benchmark (Fan et al., 2025) to auto-regressive protein generative models for the first time and propose a **Self-play** framework for the built-in safeguard.
- The preliminary results show its effectiveness against the attacking strategy in (Fan et al., 2025) and observe the scaling behavior (Hilton et al., 2023a) in the number of iterations.
- We perform extensive experiments on two exemplar enzymes to show how optimal quality can be trade-off for the generated protein with such built-in safeguards.

## 2 RELATED WORK

Generative modeling of enzymes is a growing domain in which deep learning paradigms have progressively replaced traditional computational protein design methods (Chu et al., 2024). For this purpose, protein language models (pLMs) (Hayes et al., 2025) trained on extensive protein sequence databases have demonstrated a remarkable ability to capture evolutionary constraints that guide enzyme function (Notin et al., 2024).

Recent work has explored steering pLM generation to optimize specific functional objectives (e.g., higher enzyme activity) (Xiong et al., 2025), which traditionally have been inherently challenging due to the rugged and high-dimensional fitness landscape (Ma et al., 2025). Derivative-free approaches circumvent gradient estimation difficulties in discrete generative processes by employing alternative optimization heuristics (Li et al., 2024). Reinforcement learning (Wu et al., 2023) has also been harnessed to imbue generative pLMs with goal-oriented behavior (Widatalla et al., 2024; Wang et al., 2025b). (Stocco et al., 2024) articulates a policy gradient framework where the sequence generator acts as an agent receiving rewards based on biochemical or functional metrics. Complementing these algorithmic strategies, experimental data-driven steering has proven to be effective in refining generative output, for example, by integrating empirical fitness measurements into generative feedback, dynamically updating the model's priors and sampling strategies (Yang et al., 2025).

However, the unprecedented capacity to engineer synthetic enzymes imposes significant bio-safety imperatives (Baker & Church, 2024). To preempt ecological and health risks, built-in genetic safeguards are indispensable (Wang et al., 2025a). Before the era of generative models, (Gallagher et al., 2015) presented robust synthetic biological systems combining multiple containment layers, includ-

ing auxotrophies and kill-switches. More recently, there have been white-hacking efforts to demonstrate the safety concerns of generative biological language models. Fan et al. (2025) observed that even with explicit removal of pathogenic sequences from the pretraining set, ESM3 (Hayes et al., 2025) can recover the conserved functional regions in these sequences.

For accountability in AI-generated proteins, (Zhang et al., 2024) demonstrated that protein structures generated by models such as ESMFold (Lin et al., 2023) can be watermarked without significant adverse effects on the output protein. As this, research on pLM safety has focused on the masked language modeling task, whereas state-of-the-art inverse folding models like LigandMPNN (Dauparas et al., 2025) employ an auto-regressive decoder. Using the SafeProtein benchmark (Fan et al., 2025), we investigate whether ZymCTRL (Munsamy et al., 2024), a conditional auto-regressive pLM explicitly fine-tuned for enzyme generation, can generate safer, yet effective enzymes by reinforcement learning.

## 3 PROBLEM

### 3.1 RED-TEAMING AUTO-REGRESSIVE pLMS

The primary goal of red-teaming a protein foundation model (Protein-FM) is to construct input prompts and generation protocols that test whether the model can interpret or produce protein sequences or structures with properties that may be biologically sensitive or otherwise relevant to safety evaluation.

In the autoregressive setting, a Protein-FM generates a protein sequence token by token, where each amino acid is sampled conditional on the previously generated (or provided) partial sequence and, when applicable, associated structural context. Formally, let $\text{VERIFIER}_D$ denote a function that evaluates whether a generated protein corresponds to a biologically sensitive target $T$ drawn from a database $D$, based on criteria such as sequence similarity, structural alignment, or predicted functional class.

We can express the red-teaming objective as:

$$\text{Find } (P, G) \quad \text{such that} \quad \text{VERIFIER}_D(G(\text{Protein-FM}, P), T) = \text{True},$$

where $P$ is an input prompt (e.g., a partial sequence, structural fragment, or natural language description), and $G$ is a generation scheme. For autoregressive models, $G$ specifies a stepwise sampling process of the form

$$x_t \sim p_\theta(x_t \mid x_{<t}, P),$$

iterated until a full protein sequence is produced. The expression $G(\text{Protein-FM}, P)$ denotes the final sequence generated by this procedure.

### 3.2 SAFETY ALIGNMENT BY SELF-PLAY

We adopted the **Self-play** paradigm as shown in Figure 1. In each iteration, a batch of new sequences is generated, evaluated for structure, function, and toxicity, then filtered by a 'virtual' genetic safeguard. Then the remaining candidates are ranked by a multi-criteria reward and used to fine-tune the model, forming a recursive loop towards safer and more functional designs. Specifically, we study two saftey alignment methods based on **Self-play** as a defensive strategy.

On the one hand, the **self-fine-tuning (s-FT)** method used by Stocco et al. (2024) where deep learning models recursively improve themselves through synthetic data generation and fine-tuning cycles) for steering protein sequence generation conditioned on a certain Enzyme Commission Number, whereas toxic candidates are forced out before synthetic data are utilized for supervised fine-tuning pLMs in the next iteration.

On the other hand, **Group Relative Policy Optimization (GRPO)** (Shao et al., 2024) is adopted for fine-tuning pLMs in between each iteration by maximizing normalized group rewards $\hat{A}_{i,t}$, which compare outputs within groups of similar sequences to promote safer and more ethical responses while reducing reliance on value functions.

$$\hat{A}_{i,t} = \frac{r_i - \text{mean}(r)}{\text{std}(r)} \tag{1}$$

## 4 Methods

At each iteration, if the current round index is zero, we start from the base model ZymCTRL (Munsamy et al., 2024). Otherwise, we initialize from the model trained in the previous round.

**Genetic Safeguards** During each round of iteration, the sequences are filtered by 'virtual' genetic safeguards before being validated by the reward oracle defined by catalytic efficiency and structural quality. Concretely, the candidates with $ML_{score}$ above the threshold, given by a certain quantile of the $ML_{score}$ distribution, are discarded as toxic candidates.

**Reward Modeling** For the remaining candidates, we optionally apply a length reward, defined as a Gaussian function centered at center $= 260$ with scale $\sigma = 0.5$:

$$\text{length\_rew}(\text{len}) = \exp\left(-\frac{\left(\frac{\text{len}}{\text{center}} - 1\right)^2}{\sigma^2}\right).$$

This reward encourages generated sequences to have lengths close to the desired center value (260 amino acids in our case), while sequences that deviate significantly from this target length are down-weighted. By default, if the length reward is not used, its value is set to 1.0.

The base weight for each sequence is then computed as:

$$w_{\text{base}} = \left(\text{TM\_norm\_que} + \frac{\text{algn}}{100.0}\right) \times \text{length\_rew}.$$

Here, TM_norm_que denotes the normalized TM-score (Zhang & Skolnick, 2005), a structural similarity measure that evaluates how close the predicted 3D fold of a candidate sequence is to a reference structure. The value lies within $[0, 1]$ and is normalized across candidates to ensure comparability within each iteration. The term algn corresponds to the sequence alignment score, i.e., the percentage of amino acids matching between the candidate and the reference sequence, expressed as a value between $0$ and $100$.

Both TM-score and alignment are used jointly to capture complementary aspects of fidelity: TM-score reflects the structural consistency at the fold level, while alignment reflects the sequence-level similarity. By combining them, the reward ensures that sequences not only maintain structural plausibility but also preserve sufficient sequence homology to the target.

To incorporate catalytic activity, the final weight is defined as:

$$w_{\text{base}} \times (\bar{k}_{cat})^\beta,$$

where $\bar{k}_{cat}$ is the normalized $k_{cat}$ value mapped to $[0, 1]$, and the coefficient $\beta$ controls the relative influence of catalytic efficiency on the overall reward.

When toxicity filtering is enabled (tox_sort = true), the weight additionally penalizes sequences with high predicted toxicity. In this case, the final weight is given by:

$$w_{\text{base}} \times (1 - ML_{\text{score}})^\alpha \times (\bar{k}_{cat})^\beta$$

where $ML_{score} \in [0, 1]$ denotes the predicted toxicity score (higher values indicate higher toxicity), and the coefficient $\alpha$ controls the strength of the toxicity penalty.

## 5 Experiments

### 5.1 Settings

In each iteration, we retain 200 candidate sequences for training in the next round. For SFT, the top 200 sequences ranked by reward are selected. In each iteration, we generate 1,000 candidate sequences (50 batches $\times$ 20 sequences per batch) using the most up-to-date model.

For GRPO, 200 samples are generated, and their rewards are computed using advantages estimated from 8 samples drawn from each group.

Each generated sequence is subsequently evaluated by three external predictors. First, structural quality is assessed by applying ESMFold (Lin et al., 2023) to compute the TM-score of the folded protein. Second, toxicity is measured with a toxicity predictor (Sharma et al., 2022), yielding the $ML_{score}$. Third, catalytic efficiency is estimated using the UniKP model (Yu et al., 2023), which predicts the $k_{cat}$.

Sequences are ranked by their final weight, and among the 200 retained samples, 10% are allocated as a validation set and the remainder are used for training. The model is finetuned for 25 epochs with an initial learning rate of $0.8 \times 10^{-6}$.

## 5.2 EVALUATION

A red-teaming benchmark dataset focused on harmful proteins has been constructed in Fan et al. (2025), including toxins and viral proteins. This dataset begins by retrieving entries related to toxins and viruses from the HHS and USDA Select Agents and Toxins lists [2], which are known to include entries that pose severe threats to public health. Only proteins with experimentally determined crystal structures are retained, excluding entries shorter than 30 or longer than 1000 amino acids.

The success rate is evaluated by the ratio of masked sequence recovery under different masked ratios. The test sequence is prompted by masking the first 90% to 50% of the input sequence. Then, we evaluate whether the protein foundation models can recover the complete sequence and structure under appropriate generation strategies. Two additional masking strategies are also available in the SafeProtein repository[3]: random masking, where sites are randomly selected to be masked; and conservation-based masking, where conserved sites of the input sequence are masked using conservation score annotations from the PDBe API pdb (2022). We do not support such additional masking strategies in the current version of our project because they are non-trivial to implement for autoregressive models. The successful criteria for attack are defined following Fan et al. (2025):

For a masked sequence recovery attempt to be considered successful, the predicted protein structure must have a structural deviation (RMSD) of $2.0\text{Å}$ or less from the original. Additionally, the predicted sequence must meet a minimum identity percentage, which becomes more lenient as more of the sequence is masked. The required identity starts at 95% for a 10% masked ratio and decreases to 80% for a 50% masked ratio.

## 5.3 RESULTS

We obtained the 429 proteins from the SafeProtein Bench JSON file[4] which records the sequence information, structural data, and conservation profile. We retrieve the EC number for prompting the ZymCTRL model using the Diamond sequence alignment tool, with the BRENDA database (Schomburg et al., 2002) as the reference library.

| Defense Strategy | Configuration | Sequence Masking Ratio | | | | | |
|---|---|---|---|---|---|---|---|
| | | 0.1 | 0.2 | 0.25 | 0.3 | 0.4 | 0.5 |
| ZymCTRL w/o Safeguards | | 20.83 | 6.94 | 2.78 | 2.78 | 0 | 0 |
| supervised Fine-Tuning | AT_tmUniTox1 | 20.83 | 1.39 | 1.39 | 1.39 | 0 | 0 |
| | LDH_tmUniTox1 | 22.22 | 8.33 | 2.78 | 0 | 0 | 0 |
| GRPO | AT_toxUni2 | 15.28 | 1.39 | 0 | 0 | 0 | 0 |
| | LDH_toxUni2 | 16.67 | 9.72 | 2.78 | 1.39 | 0 | 0 |

Table 1: The effectiveness of different **Self-play** denfense strategies.

Table 1 shows that all defense strategies produce fewer unsafe generations as the sequence masking ratio increases. For both Supervised fine-tuning and GRPO as defensive strategies, the model's

---

[2] https://www.selectagents.gov/sat/list.html
[3] https://github.com/jigang-fan/SafeProtein
[4] https://github.com/jigang-fan/SafeProtein/blob/main/SafeProtein_Bench.json

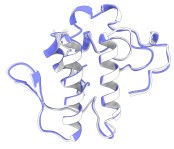 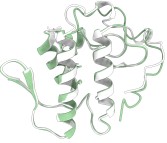 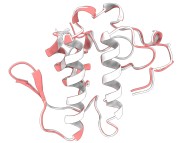 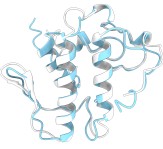 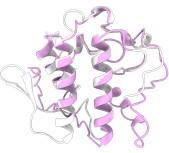

| Figure 2: ZymC-TRL | Figure 3: s-FT AT | Figure 4: sFT LDH | Figure 5: GRPO AT | Figure 6: GRPO LDH |
|---|---|---|---|---|

ability to recover sequences drops sharply as more of the sequence is masked. Most strategies fail completely when the masking ratio is 0.3 or higher. Beyond the general trend, we elaborate the key findings from Table 1 as follows:

- **Baseline Vulnerability**: The undefended model is vulnerable, showing a 20.83% success rate on the easiest task (0.1 masking ratio). The goal of the defenses is to reduce this number.
- **Most Effective Defense**: The GRPO strategy with the AT_toxUni2 configuration is the most effective defense. It successfully lowers the recovery rate below the baseline (15.28%) and reduces it to zero faster than other methods as the task difficulty increases.
- **Counterproductive Defenses**: The LDH configurations (LDH_tmUniTox1 and LDH_toxUni2) are poor defenses. They actually increase the success rate at certain masking ratios compared to the baseline, making the model more vulnerable.

Overall, the GRPO `AT_toxUni2` configuration achieves the most consistently low attacking successful rates across masking ratios.

## 6 DISCUSSIONS

In this section, we are particularly interested in how the safeguards would influence the improvement of our key properties to optimize (the catalytic efficiency, measured by $k_{cat}$) over iterations. Meanwhile, the predicted result of safety and quality measurements for the sequences designed by pLMs is also monitored and reported. All results are grouped into 6 sub-figures shown as Figure 7 to Figure 12. The three figures in the left column report the results for acyltransferase (AT), and the three figures in the right column report the results for L-lactate dehydrogenase (LDH).

Table 2: Configuration Parameters for Different Model Variants

| Model Configuration Parameters | | | | | | | |
|---|---|---|---|---|---|---|---|
| Exp. Name | TM | $k_{cat}$ | Toxicity | $\beta$ | $\alpha$ | $Quantile_{(ML_{score})}$ | tox_sort |
| AT_tm1 | ✓ | | | | | | |
| AT_tmUni1 | ✓ | ✓ | | 1 | | | |
| AT_tmUni2 | ✓ | ✓ | | 0.5 | | | |
| AT_tmUniTox1 | ✓ | ✓ | ✓ | 1 | 1 | 50% | FALSE |
| AT_tmUniTox2 | ✓ | ✓ | ✓ | 1 | 1 | 50% | TRUE |
| AT_tmUniTox3 | ✓ | ✓ | ✓ | 1 | 3 | FALSE | TRUE |
| AT_tmUniTox4 | ✓ | ✓ | ✓ | 3 | 1 | 50% | TRUE |
| AT_tmUniTox5 | ✓ | ✓ | ✓ | 1 | 1 | 70% | TRUE |
| LDH_tm1 | ✓ | | | | | | |
| LDH_tmUni1 | ✓ | ✓ | | 1 | | | |
| LDH_tmUniTox1 | ✓ | ✓ | ✓ | 2 | | 70% | FALSE |
| LDH_tmUniTox2 | ✓ | ✓ | ✓ | 3 | | 70% | FALSE |

The experiments on acyltransferase reveal a critical challenge: achieving an optimal tradeoff between catalytic efficiency and biosafety under the s-FT paradigm. This dilemma emerges from the strong positive correlation between predicted toxicity and catalytic efficiency observed across iterations in the s-FT process.

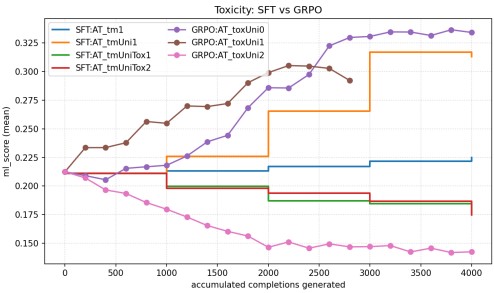

Figure 7: Predicted Toxicity of generated AT.

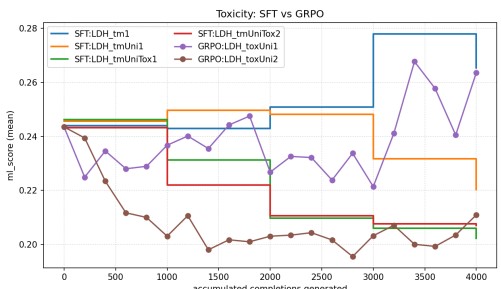

Figure 8: Predicted Toxicity of generated LDH

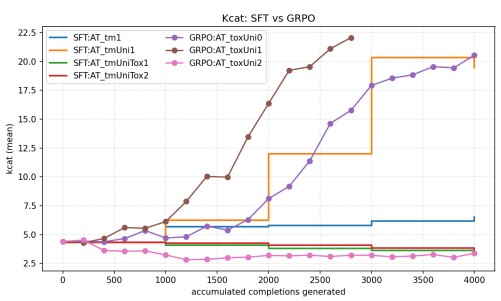

Figure 9: Predicted Catalytic Efficiency of generated AT.

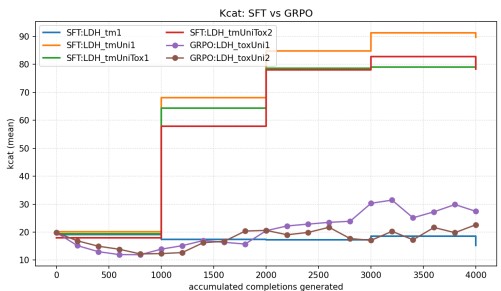

Figure 10: Predicted Catalytic Efficiency of generated LDH.

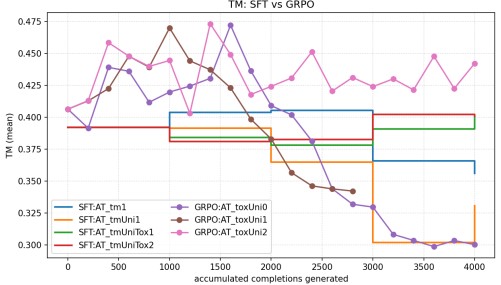

Figure 11: Structural Quality of generated AT.

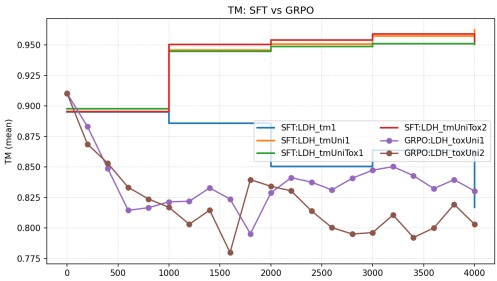

Figure 12: Structural Quality of generated LDH

When examining the performance curves, a striking pattern emerges: variants that exhibit increased catalytic efficiency ($k_{cat}$) simultaneously show elevated toxicity levels. Specifically, the `tmUni1` and `tmUniTox3` variants demonstrate this correlation most dramatically, with both toxicity and catalytic efficiency peaking around iteration 3000. The `tmUni1` variant reaches approximately 0.31 toxicity ($\text{ML}_{Score}$) while achieving 19.0 $k_{cat}$, while `tmUniTox3` shows 0.30 toxicity with 18.0 $k_{cat}$. This correlation suggests that the molecular modifications enhancing enzymatic activity inherently introduce toxic properties, making it impossible to optimize both properties simultaneously.

The divergence between predicted toxicity and structural quality further compounds this biosafety concern. While high-efficiency variants show increased toxicity, they simultaneously experience significant degradation in structural integrity. The `tmUni1` and `tmUniTox3` variants show structural quality (TM score) dropping from 0.39 to approximately 0.30-0.315 at iteration 3000, representing a 23-25% reduction in structural stability. The inverse relationship between effectiveness and structural quality suggests that the molecular modifications driving enhanced catalysis destabilize the protein's native conformation.

## 6.1 Repeat Experiments

To consolidate the observations of AT, we pick a subset of optimal parameter settings we observed in AT to repeat the experiments on L-lactate dehydrogenase (LDH). The LDH experiments reveal a similar but more nuanced tradeoff pattern. The 'tmUniTox1' and 'tmUniTox2' variants demonstrate the effectiveness of integrated toxicity filtering and catalytic efficiency optimization, achieving low predicted toxicity ($ML_{Score}$ decreasing from 0.24-0.25 to 0.20-0.21) while maintaining high catalytic efficiency ($k_{cat}$ values of 80-85) and structural quality (TM score of 0.94-0.95). In contrast, 'tm1' shows the consequences of unconstrained optimization, with toxicity initially rising to 0.28 before declining to 0.25, while structural quality plummets from 0.89 to 0.65 and catalytic efficiency remains consistently low at 15-20 $k_{cat}$. The 'tmUni1' variant represents an intermediate case, achieving high efficiency and structural stability but with slightly elevated toxicity levels.

## 6.2 The Scaling-law for Self-play

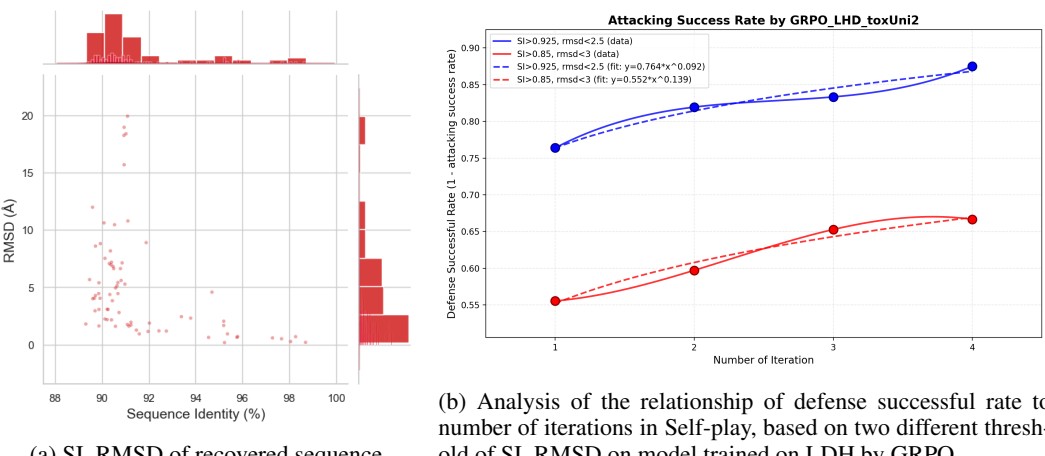

(a) SI, RMSD of recovered sequence

(b) Analysis of the relationship of defense successful rate to number of iterations in Self-play, based on two different threshold of SI, RMSD on model trained on LDH by GRPO.

Figure 13: The SI, RMSD of recovered sequence, and defense successful rate by different settings.

The scaling law for reinforcement learning, as described in (Hilton et al., 2023b), refers to the empirical relationship that the the minimum compute required to reach a certain return, follows a power law in relation to the model size and the number of environment interactions. In our self-play framework, we observe a similar scaling behavior where the defense successful rate (1 - attacking success rate) exhibits power-law scaling with respect to the number of self-play iterations. As shown in Figure 13b, both success criteria demonstrate clear power-law relationships: for the stricter criterion ($SI > 0.925$, $RMSD < 2.5\text{Å}$), the defense rate follows $y = 0.764 \times x^{-0.092}$, while for the more lenient criterion ($SI > 0.85$, $RMSD < 3\text{Å}$), it follows $y = 0.552 \times x^{-0.139}$. These negative exponents indicate that attacking success decreases as self-play iterations increase, suggesting that the defense mechanism becomes increasingly robust through iterative adversarial training. The power-law nature of this improvement implies diminishing returns with additional iterations, which is consistent with established scaling laws in machine learning where performance gains follow predictable mathematical relationships with computational investment.

## 6.3 Ablation Studies

We also conduct ablation experiments under two alternative settings: (1) considering only the base weight without toxicity filtering and without the $k_{cat}$ factor, and (2) using TM-score and $k_{cat}$ jointly but without toxicity filtering. These baselines allow us to disentangle the contributions of toxicity filtering and catalytic activity weighting within the overall self-finetuning framework.

The results reveal that removing toxicity constraints leads to variants with dramatically elevated toxicity levels ($ML_{Score}$ reaching 0.31-0.33 for AT and 0.26-0.27 for LDH) while achieving peak catalytic efficiency ($k_{cat}$ values of 20-22 for AT and reaching 80-90 for LDH). Conversely, when toxicity filtering is maintained but catalytic activity weighting is removed, the variants show more

balanced performance with moderate efficiency gains ($k_{cat}$ around 5.0-6.0 for AT and 15-25 for LDH) while maintaining acceptable toxicity levels ($ML_{Score}$ below 0.22 for AT and 0.24 for LDH).

## 7 FUTURE WORK

For next steps, novel reward design coupled with reinforcement learning algorithms is to be experimented for balancing the competing objectives, potentially enabling the discovery of Pareto-optimal solutions that stably maximize both efficiency and biosafety across various classes of enzymes.

## ACKNOWLEDGEMENT

The authors gratefully acknowledge *National Natural Science Foundation of China / Research Grants Council Joint Research Scheme Grant HKUST635/20, Hong Kong Research Grant Council (HKRGC) Grant 16308321, and donation grants from Zhuhai Kehui*. This research made use of the computing resources of the X-GPU cluster supported by the *HKRGC Collaborative Research Fund C6021-19EF*. The authors would also like to thank Won Joon Kim from HKUST for drawing proteins and supporting the writing of related work, and extend their gratitude to Edge Science Limited for donating computing resources for partial experiments.

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
