# OpenReview forum: "Built-in Safeguards for Auto-regressive Protein Foundation Models through Self-play"
_ICLR.cc/2026/Workshop/FM4Science — ICLR 2026 Workshop FM4Science Poster_

### Official Review · Reviewer_fvh8 · 2026-02-13
**Assessing Built-in Safeguards for Autoregressive Protein Models via GRPO Self-Play - Good Paper, Accept**

**Rating:** 7
**Confidence:** 3

**Review:**

This paper addresses the issue of dual-use biorisk in protein generative models. It specifically targets the gap in defensive frameworks for autoregressive models, which have been less explored compared to diffusion models. The authors propose a self-play framework to instill "virtual" genetic safeguards directly into autoregressive protein foundation models. The process involves generating sequences, evaluating them for structure, function, and toxicity, and then filtering out "toxic" candidates before using the remaining high-quality sequences to fine-tune the model. The authors evaluate two primary strategies: s-FT and GRPO. The framework is tested using the SafeProtein benchmark, measuring a model's ability to recover harmful sequences from masked prompts. They find that GRPO offers superior efficiency and safety results, though a significant trade-off remains between maintaining high catalytic activity and ensuring complete non-toxicity.

Strengths:

S1: Novel Autoregressive Focus: Addresses an important gap by extending biosecurity red-teaming and defensive frameworks to autoregressive protein models, whereas previous work focused on diffusion models.

S2: Algorithmic Innovation: Successfully adapts GRPO for protein safety alignment, demonstrating superior efficiency and better defensive outcomes compared to standard fine-tuning.

S3: Empirical Scaling Law: Establishes a power-law relationship between the number of self-play iterations and defense success, providing a predictable mathematical framework for computational safety gains.

S4: Multi-Objective Reward Design: Implements a robust reward system that simultaneously evaluates structural quality sequence alignment, and catalytic efficiency.

S5: Comprehensive Evaluation: Utilizes the rigorous SafeProtein benchmark and diverse external predictors (ESMFold, UniKP) to validate safety across different enzyme classes.

Weaknesses:

W1: Performance-Safety Conflict: Demonstrates a critical "inverse relationship" where enhancing catalytic efficiency inherently increases predicted toxicity and reduces structural stability. (This is, to me, the key weakness to address)

W2: Structural Degradation: High-efficiency variants suffered a 23–25% reduction in structural integrity (TM-score) over iterations, suggesting that safety-driven modifications may destabilize native protein folds.

W3: Counterproductive Defenses: Certain configurations (specifically LDH variants) actually increased the model's vulnerability compared to the undefended baseline at specific masking ratios.

W4: Evaluation Limitations: Does not currently support random or conservation-based masking due to implementation difficulties with autoregressive models, limiting the robustness of the red-teaming assessment.

W5: Dependency on External Oracles: The framework's reliability is strictly bound to the accuracy of external predictors like ESMFold and UniKP; any biases in these "virtual oracles" will propagate into the final model.

W6: Diminishing Returns: While following a scaling law, the safety improvements exhibit power-law behavior, implying that significantly more computational investment is required for marginal safety gains in later iterations.

I recommend to accept the paper (7), due to its pioneering work on autoregressive safeguards and its rigorous use of the SafeProtein benchmark.  Addressing the pareto-optimality of the safety-efficiency trade-off more directly, perhaps by testing more sophisticated reward-shaping techniques mentioned in the future work, could lead to further improvement.

As a little nitpick, please consider defining ML score earlier - it is used half a page before it is formally introduced.

---

### Official Review · Reviewer_F7A7 · 2026-02-19
**Auto-Regressive Protein Foundation Models through Self-Play**

**Rating:** 6
**Confidence:** 4

**Review:**

Summary:
This paper extends the SafeProtein red teaming benchmark to autoregressive protein language models and proposes a self play framework for safety alignment. GRPO and supervised fine tuning are applied to ZymCTRL, with results showing GRPO with toxicity aware rewards reduces masked sequence recovery attack success rates on hazardous proteins. A preliminary scaling observation is reported across two enzymes (AT and LDH).

Workshop Fit:
The paper addresses biosecurity for autoregressive protein foundation models, which is directly relevant to FM4Science's safety themes. The preliminary nature of the work suits the workshop format, and the failure modes reported would generate productive discussion among attendees working on protein generative models.

Strength
1) Timely and important problem. Extending safety frameworks to autoregressive protein models fills a legitimate gap with clear biosecurity relevance.
2) Practical framework. The generate, filter, fine tune self play loop is well motivated and clearly described.
3) Honest failure mode reporting. The authors candidly report counterproductive LDH configurations and the toxicity efficiency correlation, which are themselves useful findings for the community.

Weaknesses
1) The self play loop depends on ESMFold, ToxinPred2, and UniKP, all of which have known limitations in novel sequence regimes. Since the framework optimizes directly against these proxies, reward hacking and systematic bias cannot be ruled out. The observed toxicity efficiency correlation may itself be a predictor artifact, yet this is not discussed. For a workshop paper, acknowledging this limitation explicitly and discussing the likely direction of bias would strengthen the work.

2)  Table 2 lists 12 configurations but has no companion results table. Quantitative claims in the text appear to be read from figures rather than reported in verifiable form. A structured table reporting key metrics (MLscore, kcat, TM score, attack success rate) per configuration would significantly improve clarity.

3) The results would be strengthened by error bars or confidence intervals where feasible, particularly for key comparisons


3) The toxicity efficiency correlation and counterproductive LDH configurations raise interesting questions about why GRPO outperforms s FT and whether the observed trade offs are fundamental or configuration specific.

---

### Official Review · Reviewer_MKWt · 2026-02-21

**Rating:** 7
**Confidence:** 4

**Review:**

**Summary:**

This paper introduces built-in safety mechanisms into autoregressive protein foundation models. Based on the SafeProtein red-teaming benchmark, the authors adapt the attack framework to autoregressive sequence generators and propose a self-play–based defensive strategy. In each iteration, the usage of both supervised fine-tuning and GRPO reinforcement learning is discussed to update the model. Experiments are conducted on SafeProtein for attack success rate, and on two example enzymes to assess the trade-off between catalytic efficiency, toxicity, and structural quality.

**Strengths:**

1. The problem setting is well clarified, and the self-play framework is clearly structured. The comparison of SFT and GRPO RL methods adds useful insight.

2. In addition to attack success rate, this paper also examines how safety constraints interact with catalytic efficiency and structural quality.

**Weaknesses:**

1. The self-play framework incorporates ESMFold, MLscore, and UniKP external surrogate predictors for the optimization and filtering process. The paper does not discuss uncertainty, bias, or error propagation from these components. If these predictors are inaccurate, the self-play loop may optimize toward shifted surrogate-model predictions rather than true biological safety.

2. Figure 13 shows a power-law relationship between defense success rate and the number of self-play iterations, but this is based on a single model and setting. With limited data points and no variation in model size or training scale, the evidence looks more like an empirical scaling characterization than a robust scaling law in the usual sense.

3. Although the proposed approach is framed as a safeguard for protein foundation models, experiments are conducted only on ZymCTRL. The authors didn't demonstrate that the framework is model-agnostic and can generalize across different autoregressive protein models.

4. Each iteration of the self-play framework involves generating many sequences, running three external predictors, and performing fine-tuning. If multiple iterations are required, the overhead may be substantial. It would be helpful to discuss the computational considerations for practical deployment.

5. In line 146, the authors named the self-fine-tuning method as (s-FT), but there is also supervised fine-tuning in the manuscript. Can authors clarify the meaning of the acronym s-FT? And it seems that there is an additional parenthesis in line 147.

6. Only equation (1) is numbered. It would improve readability to number other key formulas and refer to them consistently in the text. Also, the caption of Table 1 should be above the table. Figures 2–6 and Figures 7–12 could be reorganized into two multi-panel figures, respectively, with clear subfigure labels.

---

### Meta-Review · Area_Chair_4zc7 · 2026-02-28

**Recommendation:** Accept (Poster)
**Confidence:** 3

**Metareview:**

This paper explores the problem of systematic red-teaming of autoregressive protein foundation models (PFMs), to help safeguard against the dual-use nature of PFMs. Reviewers found the main contribution - a model self-play defense strategy - to be timely and important for the community. The evaluation methodology is rigorous and effective at validating the main claims. A key weakness raised by reviewers is the over-reliance on external predictors (e.g., ESMFold), which raises risks of reward hacking and error propagation. A deeper investigation into the safety-efficiency trade-off is recommended.

---

### Decision · Program_Chairs · 2026-03-03

Accept (Poster)